# Chemical logic of MraY inhibition by antibacterial nucleoside natural products

Ellene H. Mashalidis[1], Benjamin Kaeser[1], Yuma Terasawa[2], Akira Katsuyama [2], Do-Yeon Kwon[3], Kiyoun Lee[4], Jiyong Hong [3], Satoshi Ichikawa[2] & Seok-Yong Lee [1]

Novel antibacterial agents are needed to address the emergence of global antibiotic resistance. MraY is a promising candidate for antibiotic development because it is the target of five classes of naturally occurring nucleoside inhibitors with potent antibacterial activity. Although these natural products share a common uridine moiety, their core structures vary substantially and they exhibit different activity profiles. An incomplete understanding of the structural and mechanistic basis of MraY inhibition has hindered the translation of these compounds to the clinic. Here we present crystal structures of MraY in complex with representative members of the liposidomycin/caprazamycin, capuramycin, and mureidomycin classes of nucleoside inhibitors. Our structures reveal cryptic druggable hot spots in the shallow inhibitor binding site of MraY that were not previously appreciated. Structural analyses of nucleoside inhibitor binding provide insights into the chemical logic of MraY inhibition, which can guide novel approaches to MraY-targeted antibiotic design.

[1] Department of Biochemistry, Duke University Medical Center, 303 Research Drive, Durham, NC 27710, USA. [2] Faculty of Pharmaceutical Sciences, Hokkaido University, Kita-12, Nihi-6, Kita-ku, Sapporo 060-0812, Japan. [3] Department of Chemistry, Duke University, Durham, NC 27708, USA. [4] Department of Chemistry, The Catholic University of Korea, Bucheon 14662, Korea. Correspondence and requests for materials should be addressed to S.-Y.L. (email: seok-yong.lee@duke.edu)

D rug-resistant bacterial infections have claimed the lives of millions of people worldwide[1], underscoring an urgent need for the development of antibacterial compounds with novel mechanisms of action. Peptidoglycan biosynthesis is a pathway rich in antibiotic targets, including the penicillin-binding proteins, which are implicated in resistance mechanisms widely documented and studied[2]. An attractive alternative and under-explored target in peptidoglycan biosynthesis is phospho-MurNAc-pentapeptide translocase (MraY), which is an essential integral membrane enzyme that catalyzes the first membrane-associated and committed step of peptidoglycan formation[3–5]. MraY transfers phospho-MurNAc-pentapeptide from the hydrophilic substrate uridine diphosphate-MurNAc-pentapeptide (UM5A), to the lipid carrier undecaprenyl phosphate ($C_{55}$-P) in the presence of a $Mg^{2+}$ cofactor. The resulting product is Lipid I, an intermediate in peptidoglycan biosynthesis (Supplementary Fig. 1a).

MraY is the target of five classes of natural product nucleoside inhibitors isolated from *Streptomyces* species with promising activity against pathogenic bacteria: the liposidomycins/caprazamycins, capuramycins, mureidomycins, muraymycins, and tunicamycins. Each MraY inhibitor contains a uridine moiety, but they otherwise differ in their core chemical structures. Nucleoside natural product inhibitors exhibit differing antibacterial activity, structure-activity-relationship (SAR) profiles[6,7], mechanisms of action[8,9], and inhibitor kinetics[8–10] Tunicamycin inhibits both MraY and its eukaryotic paralog GlcNAc-1-P-transferase (GPT), leading to cytotoxicity[11], but members of the other classes of nucleoside inhibitors are selective for bacterial MraY[9,12]. The mechanistic and structural basis for the distinct pharmacological properties observed among MraY-targeted nucleoside inhibitors is poorly understood.

Recent structures of tunicamycin bound to MraY[13] and GPT[14,15] show that the tunicamycin binding pocket is deep and occluded in GPT, while in MraY it is shallow and largely exposed to the cytosol. The MraY inhibitor binding site on the cytoplasmic face of MraY is unlike the large, deep, and enclosed binding pockets typically found in enzyme active sites[16]. This observation raises an intriguing and important question: what strategy does nature employ to target the shallow cytosolic surface of MraY using nucleoside inhibitors with very different core chemical structures? One possibility is that the structural plasticity of MraY helps to accommodate structurally diverse inhibitors, as suggested by comparison of apoenzyme and muraymycin D2-bound MraY[17,18]. Alternatively, it is possible that the shallow surface of MraY contains many cryptic druggable sites, which can be exploited in different combinations by each nucleoside inhibitor. To address this question, we solved structures of MraY from *Aquifex aeolicus* (MraY_AA) individually bound to carbacaprazamycin (a member of the caprazamycin class), capuramycin, and 3′-hydroxymureidomycin A (a ribose derivative of mureidomycin A). These three classes of nucleoside inhibitors are distinct in their chemical structures, mechanisms of inhibition, and antibacterial activity. For example, liposidomycin is competitive for $C_{55}$-P, the lipid carrier substrate of MraY[8], while capuramycin is non-competitive for $C_{55}$-P and exhibits mixed type inhibition with respect to UM5A[9]. The liposidomycins/caprazamycins demonstrate potent antibacterial activity against Gram-positive bacteria, mycobacteria, and various drug-resistant bacterial strains, including MRSA and VRE[19]. Mureidomycin and its analogs appear to have a narrower spectrum of activity, primarily against *Pseudomonas* species[20,21], while the capuramycins are particularly effective against mycobacteria[22,23]; capuramycin analog SQ641 kills *Mycobacterium tuberculosis* faster than existing antitubercular drugs[24].

Our structures cover the chemical space sampled by MraY natural product inhibitors, revealing that they occupy both overlapping and unique sites on the cytoplasmic surface of MraY. This region of MraY is highly conserved among Gram-positive and Gram-negative bacteria, with 34 invariant amino acid residues comprising the active site[17,18]. Therefore, our crystal structures collectively serve as a generalizable MraY structural model by which nucleoside inhibitor SAR data can be analyzed and understood in order to achieve a comprehensive picture of MraY inhibition.

## Results

**Crystal structures of MraY bound to nucleoside inhibitors**. We previously identified a biochemically stable ortholog of MraY from thermophile *Aquifex aeolicus* (MraY_AA), with which we obtained crystal structures of MraY in its apoenzyme form[17] as well as bound to muraymycin D2[18]. MraY_AA is a good model with which to study MraY activity and inhibition because it recognizes the same substrates and catalyzes the same enzymatic reaction as do pathogenic Gram-positive and Gram-negative bacteria[17]. MraY_AA enzymatic activity is potently inhibited by carbacaprazamycin, capuramycin, and 3′-hydroxymureidomycin A with IC_{50} values of 104 nM, 185 nM, and 52 nM, respectively (Supplementary Fig. 1b), as well as by muraymycin D2[18] and tunicamycin[14], which is comparable to the efficacy observed for MraY orthologs from pathogenic bacteria[8–10,12,25–29]. MraY_AA was recalcitrant to crystallization in complex with members of the liposidomycin/caprazamycin, capuramycin, and mureidomycin classes of MraY inhibitors using previously employed methods. We addressed this challenge by obtaining different crystal forms of MraY_AA in the presence of camelid single-chain antibodies called nanobodies. We identified several high-affinity MraY_AA nanobodies that bind MraY_AA, forming a complex that remained intact during size exclusion chromatography. One nanobody in particular, NB7, forms a tight complex with MraY_AA (Supplementary Fig. 2a), but does not interfere with MraY_AA activity and inhibition (Supplementary Fig. 2b). The ternary complex crystals of NB7, MraY_AA, and either carbacaprazamycin, capuramycin, or 3′-hydroxymureidomycin A diffract to 2.95 Å, 3.62 Å, and 3.70 Å resolutions, respectively (Fig. 1). MraY_AA crystallizes as a dimer, which is consistent with its oligomeric state[17]. NB7 binds to each MraY_AA protomer on its periplasmic face, away from the catalytic and inhibitor binding site (Supplementary Fig. 2c). Phasing was obtained by molecular replacement and models were refined to good geometry and statistics (Table 1). Two MraY_AA-NB7 dimer complexes are found in the asymmetric unit with inhibitor density strongest in one of the MraY_AA protomers. The electron density maps for the structures were of high quality, allowing unambiguous placement of each inhibitor (Supplementary Fig. 3).

Each MraY inhibitor binds to a site on the cytoplasmic face of MraY, which is predominantly formed by TMs 5, 8, and 9b and Loops C, D, and E (Fig. 1 and Supplementary Fig. 4). Our three new crystal structures of MraY_AA bound to carbacaprazamycin, capuramycin, and 3′-hydroxymureidomycin A reveal that the conformations of inhibitor-bound MraY structures are more similar to the MraY-muraymycin D2 complex than they are to the apoenzyme MraY structure (Supplementary Fig. 5). The degree of TM9b bending and the structure of Loop E vary in each inhibitor-bound structure (Supplementary Fig. 5).

**The carbacaprazamycin binding site**. Carbacaprazamycin is a chemically stable analog of naturally occurring caprazamycin, which is a member of the lipopeptidyl class of MraY nucleoside inhibitors that includes the liposidomycins[26]. Carbacaprazamycin is comprised of uridine, 5-aminoribosyl, diazepanone, and

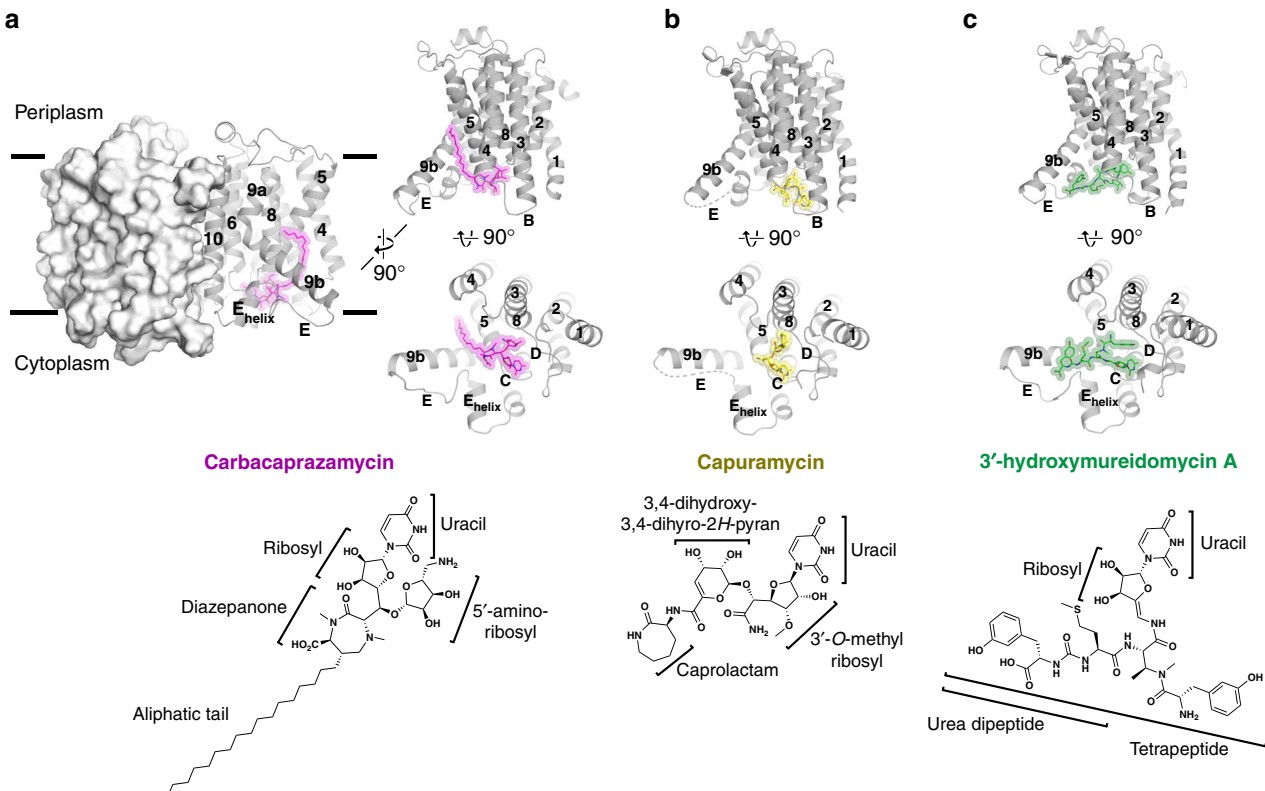

**Fig. 1** X-ray crystal structures of MraY$_{AA}$ bound to nucleoside inhibitors. **a** Top: the MraY$_{AA}$-carbacaprazamycin complex structure as viewed from the membrane, with one protomer shown in surface representation and one in cartoon. Carbacaprazamycin is shown in magenta. For simplicity, one protomer of MraY$_{AA}$ with bound carbacaprazamycin is shown from membrane and cytoplasmic views. **a** Bottom: chemical structure of carbacaprazamycin with its substructures labeled. (**b**, top) Membrane and cytoplasmic views of an MraY$_{AA}$ protomer bound to capuramycin (yellow). Loop E in is distorted and is represented by a dashed line. **b** Bottom: chemical structure of capuramycin. **c** Top: membrane and cytoplasmic views of an MraY$_{AA}$ protomer bound to 3'-hydroxymureidomycin A (green). **c** Bottom: chemical structure of 3'-hydroxymureidomycin A. Each MraY inhibitor binds distinctly to a site on the cytoplasmic face of MraY formed by TMs 5, 8, and 9b and Loops C, D, and E (labeled throughout)

aliphatic tail moieties (Fig. 1a). These moieties bind to pockets on the cytoplasmic face of MraY we term the uridine, uridine-adjacent, TM9b/Loop E and hydrophobic pockets (Fig. 2a). The uridine binding pocket in MraY is formed by amino acid residues in Loop C, including G194, L195, and D196, and is capped off by a π–π stacking interaction with F262 in Loop D (residue numbering for MraY$_{AA}$) (Fig. 2b). An additional hydrogen bond with the uracil moiety is formed by K70. The orientation and binding mode of the carbacaprazamycin uridine moiety is very similar to that observed in the crystal structures of muraymycin D2 and tunicamycin bound to MraY$_{AA}$[18] and MraY from *Clostridium bolteae* (MraY$_{CB}$)[13], respectively (Supplementary Fig. 4). Next to the uridine binding site in MraY is a second binding pocket lined with amino acid residues T75, N190, D193, and G264, which we call the uridine-adjacent pocket. The 5-aminoribose moiety of carbacaprazamycin forms an extensive hydrogen bond network in the uridine-adjacent pocket (Fig. 2b), as does this moiety in muraymycin D2[18]. The diazepanone ring system makes relatively few interactions with the protein. This observation is consistent with SAR studies demonstrating that the diazepanone ring can be broken with modest effect on activity[30,31]. However, the carboxylate group on the diazepanone forms a hydrogen bond with H325 in the Loop E helix (Fig. 2b).

Adjacent to the highly-charged nucleoside binding pocket on the cytoplasmic side of MraY is a long hydrophobic groove predominantly formed by TMs 5 and 9b leading into the plane of the membrane, which has been predicted to be the lipid carrier substrate C$_{55}$-P binding site[17]. Two of the major nucleoside

inhibitor classes, the liposidomycins and the tunicamycins, contain aliphatic moieties that are thought to compete with the lipid carrier substrate, C$_{55}$-P[8,9]. In the previously published tunicamycin-MraY$_{CB}$ complex structure[13], the acyl tail of tunicamycin was disordered and therefore was unmodeled (Supplementary Fig. 4d). Our structure of carbacaprazamycin in complex with MraY$_{AA}$ definitively demonstrates that the acyl moiety binds to the hydrophobic groove of MraY (Fig. 2c, Supplementary Fig. 6a). SAR studies of carbacaprazamycin indicate that its aliphatic tail is critical for activity; the deacylated caprazol core on its own has no antibacterial activity[31]. The core nucleoside geometry of carbacaprazamycin provides the directionality needed to target the hydrophobic binding site with specificity. This observation is consistent with previous work showing the geometry of the lipid carrier substrate, which is thought to bind to the hydrophobic groove, is critical for MraY activity[14].

**The capuramycin binding site**. Capuramycin consists of uracil and 3-O-methyl ribosyl moieties (collectively referred to as uridine), a 3,4-dihydroxy-3,4-dihydro-2*H*-pyran moiety, and a caprolactam moiety (Fig. 1b). These moieties bind to the uridine, uridine-adjacent, and caprolactam binding sites (Fig. 3a). The uridine moiety of capuramycin binds to MraY$_{AA}$ by engaging in interactions with G194, L195, D196, and F262, as does this moiety in carbacaprazamycin, muraymycin D2, and tunicamycin (Fig. 3b). At the uridine-adjacent site, the 3,4-dihydroxy-3,4-

**Table 1 Data collection and refinement statistics**

|  | Carbacaprazamycin[a] (6OYH) | Capuramycin[b] (6OYZ) | 3′-hydroxymureidomycin A[c] (6OZ6) |
|---|---|---|---|
| Data collection |  |  |  |
| Space group | P2₁ | P2₁ | P2₁ |
| Cell dimensions |  |  |  |
| a,b,c (Å) | 93.9 129.6 129.4 | 93.8 128.1 129.5 | 94.9 130.0 130.2 |
| α,β,γ (°) | 90.0 109.3 90.0 | 90.0 110.8 90.0 | 90.0 109.4 90.0 |
| Resolution (Å) | 43.94–2.95 (3.06–2.95) | 87.23–3.62 (3.75–3.62) | 72.55–3.70 (3.83–3.70) |
| R-pim | 0.079 (0.63) | 0.16 (0.52) | 0.30 (4.89) |
| Mean I/sigma(I) | 12.86 (1.20) | 10.25 (1.37) | 3.55 (1.25) |
| CC1/2 | 0.89 (0.44) | 0.96 (0.34) | 0.97 (0.66) |
| Completeness (%) | 98.70 (91.28) | 99.95 (99.82) | 100.00(100.00) |
| Completeness (%)[d] | — | — | 91.15(68.42) |
| Ellipsoidal completeness (%) | — | — | 100.00(99.4) |
| Multiplicity | 4.3 (2.6) | 15.2 (9.6) | 15.4 (11.6) |
| Refinement |  |  |  |
| Resolution (Å) | 43.94–2.95 (3.06–2.95) | 87.64–3.62 (3.749–3.62) | 72.55–3.70 (3.832–3.70) |
| No. reflections | 60,824 (5560) | 32,752 (3238) | 29,150 (2190) |
| Rwork/Rfree (%) | 24.8/26.8 | 27.6/30.0 | 25.7/30.1 |
| Number of non-hydrogen atoms | 13,851 | 12,944 | 13,180 |
| Macromolecules | 13,611 | 12,904 | 13,000 |
| Ligands | 224 | 40 | 180 |
| Water | 16 | — | — |
| Average B-factor | 80.19 | 108.50 | 82.84 |
| Macromolecules | 79.96 | 108.42 | 82.49 |
| Ligands | 95.66 | 132.80 | 107.79 |
| Water | 60.16 | — | — |
| R.M.S deviations |  |  |  |
| Bond lengths (Å) | 0.005 | 0.005 | 0.004 |
| Bond angles (°) | 0.73 | 0.74 | 0.70 |

[a]Merged from two crystals; [b]Merged from five crystals; [c]Merged from four crystals; [d]After applying anisotropy correction with STARANISO

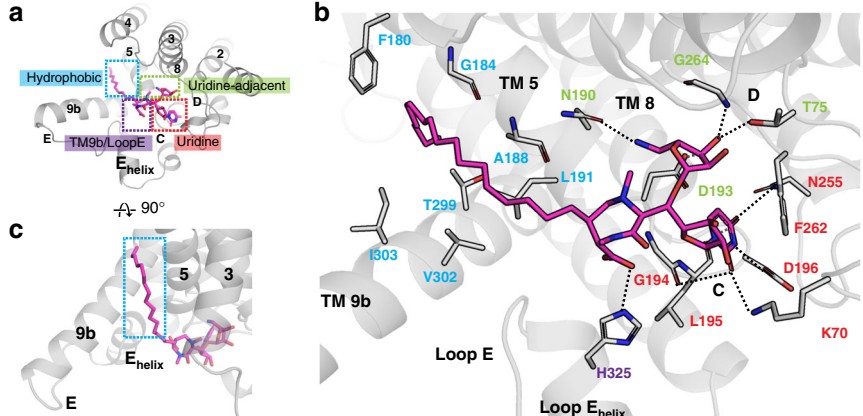

**Fig. 2** Carbacaprazamycin binds to the hydrophobic groove in MraY$_{AA}$. **a** The binding sites recognized by carbacaprazamycin (magenta) on the cytoplasmic side of MraY$_{AA}$ include the uridine (red), uridine-adjacent (lime green), TM9b/Loop E (purple), and hydrophobic (cyan) pockets. **b** A zoomed-in view of the carbacaprazamycin binding site in the same orientation as shown in **a**. Residues forming interactions with carbacaprazamycin are labeled and color-coded according to the binding pocket to which they belong. Hydrogen bonds are represented by dashed lines. **c** A view of the carbacaprazamycin binding site rotated 90° relative to the orientation shown in **a** to highlight the aliphatic tail binding site (cyan dashes lines). TMs (numbers) and Loops (letters) are labeled throughout

dihyro-2*H*-pyran moiety of capuramycin, as well as the amide linker to the caprolactam group, forms hydrogen bonds with T75, D193, and the backbone of G264 (Fig. 3b).

Notably, the caprolactam moiety of capuramycin assumes a unique binding site on the cytoplasmic face of MraY that has not been previously observed in muraymycin D2 and tunicamycin, and is unique among all MraY nucleoside inhibitors. The caprolactam occupies a very shallow, mostly hydrophobic pocket (Fig. 3c). Extensive SAR studies have been

carried out on the caprolactam moiety of capuramycin. Replacing the caprolactam moiety with a small functional group, such as a hydroxyl, amide, or methoxy group, results in dramatic loss of inhibitory activity[32]. In addition, modifying capuramycin with alkyl groups of various lengths in place of the caprolactam group reduces inhibition. However, capuramycin derivatives with cyclic functional groups instead of the caprolactam, such as phenyl, phenethyl, benzyl, cyclohexyl, and cycloheptyl moieties, have comparable activity to

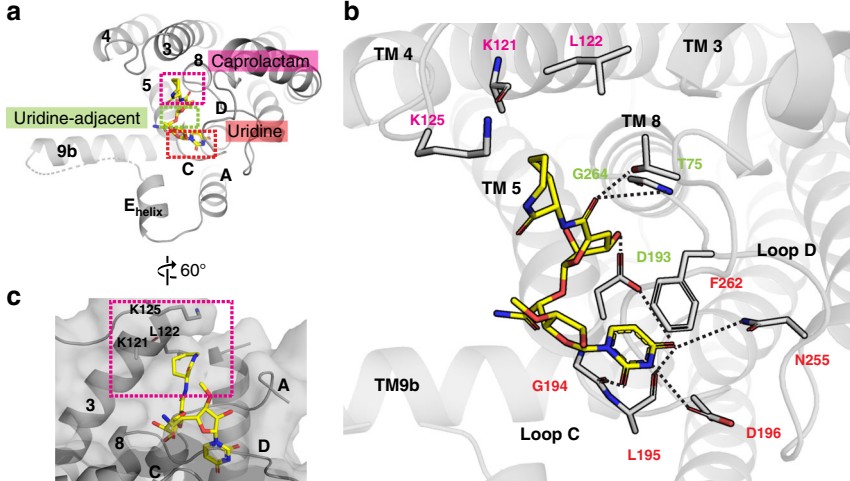

**Fig. 3** Capuramycin forms a unique interaction with the caprolactam site on MraY$_{AA}$. **a** The binding pockets occupied by capuramycin (yellow) on the cytoplasmic face of MraY$_{AA}$ include the uridine (red), uridine-adjacent (lime green), and caprolactam (pink) sites. **b** A zoomed-in view of the capuramycin binding site in the same orientation as shown in **a**. Residues forming interactions with capuramycin are labeled and color-coded according to the binding pocket to which they belong. Hydrogen bonds are represented by dashed lines. **c** A view of the capuramycin binding site rotated 60° relative to the orientation in **a** to highlight the caprolactam binding site. The surface of MraY$_{AA}$ is shown in transparent gray with residues forming the shallow caprolactam binding pocket (pink dashes lines) labeled. TMs (numbers) and Loops (letters) are labeled throughout. The side chains of residues K70 and K121 are disordered in the MraY$_{AA}$-capuramycin complex structure

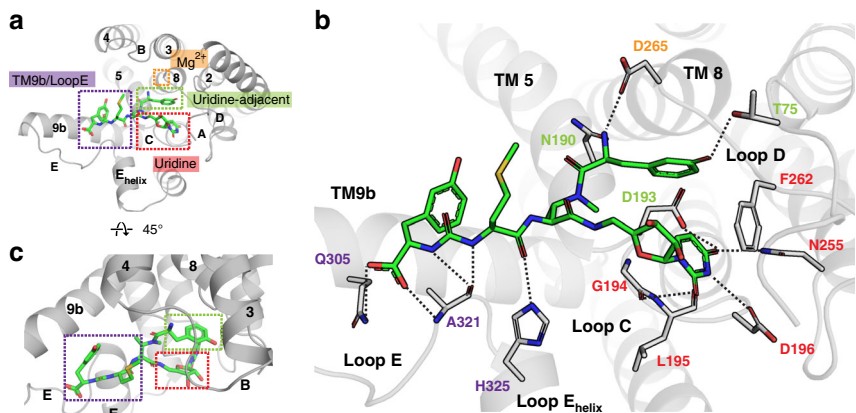

**Fig. 4** A detailed view of the interactions between MraY$_{AA}$ and 3′-hydroxymureidomycin A. **a** The binding pockets recognized by 3′-hydroxymureidomycin A (bright green) on the cytoplasmic side of MraY$_{AA}$ include the uridine (red), uridine-adjacent (lime green), TM9b/Loop E (purple), and Mg$^{2+}$ (gold) sites. **b** A zoomed-in view of the 3′-hydroxymureidomycin A binding site in the same orientation as shown in **a**. Residues forming interactions with 3′-hydroxymureidomycin A are labeled and color-coded according the binding pocket to which they belong. Hydrogen bonds are represented by dashed lines. **c** A view of the 3′-hydroxymureidomycin A binding site rotated 45° relative to the orientation shown in **a**. TMs (numbers) and Loops (letters) are labeled throughout. The side chain of residue K70 is disordered in the MraY$_{AA}$-3′-hydroxymureidomycin A complex structure

capuramycin itself[22]. These findings are consistent with the structure of the caprolactam binding pocket, which is a superficial groove (Fig. 3c); shape complementarity likely enhances affinity to the caprolactam pocket. Although part of caprolactam binding site and some nearby residues are conserved, neighboring sites are variable among MraY orthologs (Supplementary Fig. 6b). Taking into account MraY sequence variability in the regions neighboring the caprolactam binding pocket could lead to the development of capuramycin analogs with more narrow-spectrum activity.

**The 3′-hydroxymureidomycin A binding site**. The mureidomycins contain a tetrapeptide, which includes a *meta*-tyrosine moiety and an urea dipeptide motif (Fig. 1c). The tetrapeptide connects to the uridine moiety via an enamide linker. The synthetic derivative 3′-hydroxymureidomycin A,

which differs from the mureidomycins in that it contains two hydroxyl groups on the ribosyl moiety instead of one, was designed and synthesized based on a previous study[33] (Supplementary Methods). The substructures of 3′-hydroxymureidomycin A bind to the uridine, uridine-adjacent, and TM9b/Loop E pockets on the cytoplasmic side of MraY (Fig. 4a). The uridine moiety of 3′-hydroxymureidomycin A binds in a similar manner as observed for other nucleoside inhibitors, with added stabilization from a hydrogen bond with D193 (Fig. 4b). The uridine-adjacent pocket binds the *meta*-tyrosine moiety of 3′-hydroxymureidomycin A wherein T75 forms a hydrogen bond with the hydroxyl group of *meta*-tyrosine and N190 anchors the terminal amino group. This same amino group in the *meta*-tyrosine interacts with D265, the conserved and essential aspartate residue responsible for coordinating the Mg$^{2+}$ cofactor in MraY[17].

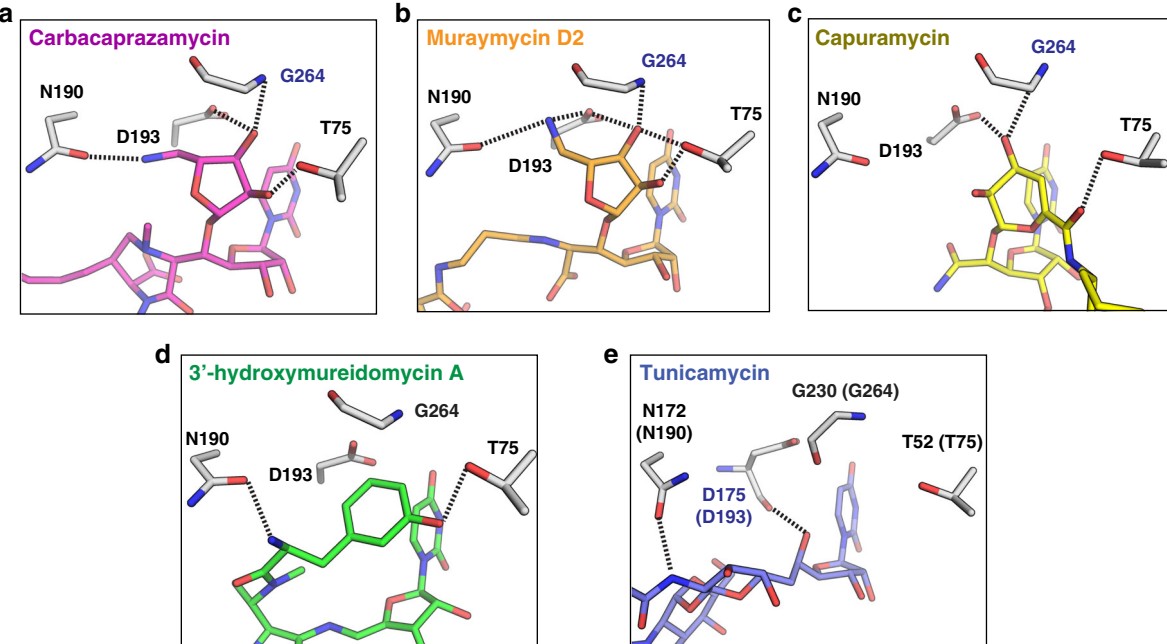

**Fig. 5** Structurally diverse moieties occupy the uridine-adjacent binding pocket in MraY$_{AA}$. **a** Carbacaprazamycin (magenta), **b** muraymycin D2 (orange), **c** capuramycin (yellow), **d** 3'-hydroxymureidomycin A (green), **e** tunicamycin (slate) binding interactions at the uridine-adjacent pocket. Residues labeled in black and blue form side chain and backbone interactions with the inhibitor, respectively. Hydrogen bonds are represented by black dashed lines. Residue numbering is shown for MraY$_{AA}$ except in **e**, which shows residue numbering for MraY$_{CB}$ with the corresponding residues in MraY$_{AA}$ provided in parentheses

The TM9b/Loop E pocket is extensively occupied by 3'-hydroxymureidomycin A (Fig. 4c). The urea dipeptide motif in 3'-hydroxymureidomycin A interacts with the TM9b/Loop E pocket by engaging Q305, A321, and H325. Muraymycin D2 also includes a urea dipeptide motif, although the identities of the amino acids in each inhibitor differ; 3'-hydroxymureidomycin A contains methionine-urea-*meta*-tyrosine, while the analogous substructure in muraymycin D2 is l-*epi*-capreomycidine-urea-valine (Supplementary Fig. 4a). Interestingly, despite the different amino acids in each compound, the urea dipeptide motif of each inhibitor binds to the TM9b/LoopE pocket similarly (Supplementary Fig. 6c).

**The uridine site is common to MraY nucleoside inhibitors.** A feature common to all MraY-inhibitor complex structures is the binding pocket that accommodates the uridine present in each nucleoside inhibitor. Structural superimposition of all five inhibitor-bound MraY crystal structures reveals that the uracil and ribosyl moieties of each nucleoside inhibitor overlap (Supplementary Fig. 7). The residues that form the uracil binding pocket of MraY (K70, G194, L195, D196, and F262) are likely involved in binding the natural substrate of this enzyme, UM5A, which also contains a uracil. There is some spatial tolerance within the uridine pocket of MraY for positioning of the uracil moiety. For example, the capuramycin and 3'-hydroxymureidomycin A uracil moieties deviate from that of the other nucleoside inhibitors (Supplementary Fig. 7). Although the uridine pocket is the most defined and enclosed pocket on the cytoplasmic surface of MraY, it is relatively accommodating to a variety of ligands containing a uracil moiety.

The ribosyl moiety of each inhibitor (or the 3'-O-methylated ribosyl, as in the case of capuramycin) also assumes a very similar orientation in each structure (Supplementary Fig. 7). No hydrogen-bonding interactions are observed between MraY and the hydroxyl groups of the ribosyl moiety in each inhibitor, which are mostly exposed to the cytoplasm. It appears that the geometry

assumed by the ribosyl moiety may provide the directionality needed for each inhibitor to occupy key binding sites on the cytoplasmic face of MraY.

**Diverse pharmacophores target the uridine-adjacent pocket.** The uridine-adjacent pocket is lined by amino acid residues T75, N190, D193, and G264. Interestingly, the spatial orientation of these residues is similar in each inhibitor-bound structure, but the pocket can accommodate very different chemical moieties (Fig. 5). Carbacaprazamycin and muraymycin D2 each contain a 5-aminoribosyl moiety, which occupies the uridine-adjacent pocket (Fig. 5a–b). The amino group of the 5-aminoribosyl moiety in carbacaprazamycin and in muraymycin D2 forms a critical interaction with D193 in the uridine-adjacent pocket. SAR studies on the 5-aminoribose-nucleoside core shared by muraymycin and carbacaprazamycin demonstrate that replacing the amino group in the 5-aminoribosyl moiety dramatically reduces inhibitory activity[34]. Furthermore, mutagenesis studies on MraY$_{AA}$ show that a D193N mutation nearly completely abolishes its affinity to muraymycin D2[18].

Capuramycin binds the uridine-adjacent pocket with a 3,4-dihydroxy-3,4-dihyro-2*H*-pyran moiety (Fig. 5c). One hydroxyl group of the 3,4-dihydroxy-3,4-dihyro-2*H*-pyran moiety engages in a hydrogen-bonding interaction with D193 and the backbone amino group of G264. Replacing this hydroxyl group with a hydrogen leads to an order of magnitude decrease in capuramycin inhibitory activity[9]. In the 3'-hydroxymureidomycin A-bound structure of MraY, the *meta*-tyrosine moiety engages in interactions with N190 and T75 in the uridine-adjacent pocket (Fig. 5d). The mureidomycin class of MraY nucleoside inhibitors belongs to a larger group of structurally-similar uridylpeptide compounds, including the pacidamycins and napsamycins, which differ with respect to the types of amino acid residues found in the peptidic moiety of each inhibitor[6]. In place of the *meta*-tyrosine moiety found in the mureidomycins, the napsamycins

have an unusual bicyclic amino acid residue, which contains *meta*-tyrosine, and some analogs of pacidamycin contain an alanine residue at the analogous position. Mureidomycin, pacidamycin, and napsamycins exhibit similar activity[35,36] and it is likely that the uridine-adjacent site accommodates the various amino acid residues found in each uridylpeptide subclass.

Compared to the extensive hydrogen-bonding networks formed by carbacaprazamycin, muraymycin D2, capuramycin, and 3′-hydroxymureidomycin A in the uridine-adjacent pocket, tunicamycin makes relatively few interactions at that site (Fig. 5e). Instead, the tunicamine sugar moiety of tunicamycin picks up hydrogen bonds with additional residues near the uridine-adjacent pocket, including K133 (K111 in MraY$_{CB}$), and a backbone interaction with L191 (F173 in MraY$_{CB}$) (Supplementary Fig. 4d). These interactions are unique to tunicamycin. Tunicamycin is the only non-selective nucleoside inhibitor among the five known classes, with off-target effects on the human MraY paralog, GPT. GPT lacks a binding pocket analogous to the uridine-adjacent pocket in MraY[14,15]. Therefore, targeting the uridine-adjacent pocket could be a strategy to engineer selectivity of nucleoside inhibitors for MraY over GPT. Occupying the uridine-adjacent pocket is not required for MraY inhibition, but it enhances inhibitory potency. This observation is bolstered by SAR studies demonstrating that muraymycin analogs lacking the 5-aminoribosyl moiety that binds the uridine-adjacent site, such as muraymycins A5 and C4 and some synthetic 5′-defunctionalized muraymycin derivatives, retain inhibitory activity[37–39]. The uridine-adjacent pocket is an opportunistic site that is highly tolerant to a wide variety of pharmacophores and can greatly enhance inhibitor binding to and specificity for MraY.

**Each MraY inhibitor binds TM9b/Loop E except capuramycin.** There is variability in the degree to which each inhibitor interacts with the most structurally plastic regions of MraY, including TM9b, Loop E, and the Loop E helix. The inhibitors 3′-hydroxymureidomycin A and muraymycin D2 make the most extensive interactions at this site, forming hydrogen bonds with Q305 and A321 in TM9b via the carboxylate and urea moieties these compounds share (Supplementary Fig. 6c). In addition, 3′-hydroxymureidomycin A and muraymycin D2 form an interaction with H325 in the Loop E helix, which is also observed in carbacaprazamycin and tunicamycin binding (Supplementary Fig. 8). Muraymycin D2 interacts with H325 via a water-mediated hydrogen-bonding network, which also includes H324, and the L-*epi*-capreomycidine moiety of the inhibitor packs against H325 as well. Carbacaprazamycin, tunicamycin, and 3′-hydroxymureidomycin A engage in hydrogen bonds with H325 directly (Supplementary Fig. 8).

Capuramycin is unique among the five classes of nucleoside MraY inhibitors because it binds away from TM9b and the Loop E helix (Supplementary Fig. 8e). Consistent with this observation, Loop E in the capuramycin-bound structure is disordered, likely because inhibitor binding does not stabilize the loop. Unlike the other MraY natural product inhibitors, capuramycin is too far away from the Loop E helix to interact with H325.

**Two nucleoside inhibitors bind the Mg$^{2+}$ site.** Three aspartate residues (D117, D118, D265 in MraY$_{AA}$, termed the DDD motif) are critical for MraY enzymatic activity and have been thought to play a critical role in catalysis[17,40]. The structure of MraY$_{AA}$ in complex with its Mg$^{2+}$ cofactor reveals that D265 is the Mg$^{2+}$-coordinating residue[17]. Among the five MraY nucleoside inhibitors, only 3′-hydroxymureidomycin A and tunicamycin interact with D265 (Supplementary Fig. 9). This interaction is formed by the tunicamine sugar moiety in tunicamycin and the

amino group of the *meta*-tyrosine in 3′-hydroxymureidomycin A. Our structural observations are fully consistent with previous studies demonstrating that tunicamycin and mureidomycin compete with the Mg$^{2+}$ cofactor binding to MraY. For example, increasing MgCl$_2$ concentration decreases the equilibrium binding constant ($K_d$) of tunicamycin to MraY$_{AA}$, as measured by isothermal titration calorimetry[14]. An analogous enzymatic assay performed with MraY from *E. coli* (MraY$_{EC}$) demonstrates that the inhibitory activity of mureidomycin and analogs thereof decreases with increasing concentrations of MgCl$_2$[41]. The authors of this study proposed that the amide linkage of the meta-tyrosine moiety in mureidomycin could interact with the Mg$^{2+}$ cofactor binding site[41,42], as in fact our structure of MraY$_{AA}$ bound to 3′-hydroxymureidomycin A demonstrates (Supplementary Fig. 9). None of the five MraY nucleoside inhibitors for which X-ray crystal structures are available form interactions with the remaining two conserved aspartate residues of the DDD motif (D117 and D118 in MraY$_{AA}$) (Supplementary Fig. 9).

**The hot spots of inhibition on MraY are summarized by a "barcode".** Occupying the uridine binding pocket appears to be critical for the inhibition of MraY by nucleoside natural products, which all bind to this pocket in a similar manner (Fig. 6a). However, each nucleoside inhibitor forms different interactions with the other binding sites on the cytoplasmic face of MraY (Fig. 6a). These sites constitute the druggable hot spots (HSs) of MraY inhibition, which we name HS1–6, representing the uridine-adjacent, TM9b/Loop E, caprolactam, hydrophobic, Mg$^{2+}$ cofactor, and tunicamycin binding pockets, respectively (Fig. 6b). An analysis of interactions each inhibitor forms with the residues comprising the six HSs reveals trends among nucleoside inhibitors that provide mechanistic insight into MraY inhibition, which we have summarized for each compound with a "barcode" tool (Fig. 6c). This analysis reveals that in addition to binding the uridine pocket, each inhibitor must form substantial interactions with at least two HSs. For example, capuramycin binds HS1 and HS3, while carbacaprazamycin occupies HS1 and HS4, and forms one hydrogen bond in HS2 (Fig. 6c). Both 3′-hydroxymureidomycin A and muraymycin D2 form interactions with HS1 and HS2; however, 3′-hydroxymureidomycin A makes two fewer contacts at these sites than does muraymycin D2 and instead picks up an additional interaction in HS5. Tunicamycin has a substantially different HS binding profile than other nucleoside inhibitors (Fig. 6c). Tunicamycin makes few interactions in HS1 and HS2, likely interacts with HS4, binds HS5, and also forms hydrogen bonds in HS6, a site on MraY not exploited by other nucleoside inhibitors. Tunicamycin achieves these interactions via its tunicamine sugar and GlcNAc moieties, two pharmacophores that are recognized by eukaryotic GPT[14]. The binding signature of tunicamycin among the nucleoside natural products is consistent with its promiscuous inhibitory activity.

## Discussion
Our structures of MraY bound to naturally occurring nucleoside inhibitors provide insights into the druggability of the shallow, solvent-exposed binding site on the cytoplasmic surface of MraY. Surface binding sites have traditionally been challenging to target for drug-development[43]. Our comparative structural analyses elucidate the design principle of MraY natural product inhibitors, serving as an instructive example in which nature overcomes the challenge of targeting surface binding sites.

The only defined "pocket" on the cytoplasmic face of MraY binds the uracil moiety of each nucleoside inhibitor. However, uridine itself cannot be a strong MraY inhibitor, if at all; therefore, MraY inhibition can be achieved by also targeting hot spots

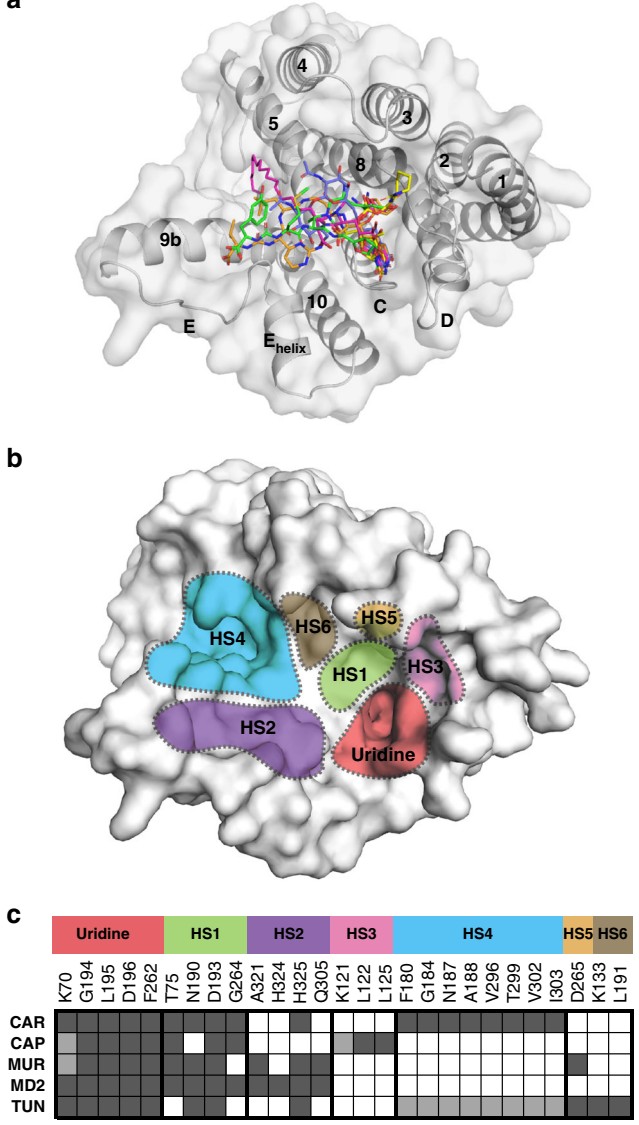

| | Uridine | | | | HS1 | | | | HS2 | | | | HS3 | | | HS4 | | | | | | | | HS5 | HS6 | |
|---|---|---|---|---|---|---|---|---|---|---|---|---|---|---|---|---|---|---|---|---|---|---|---|---|---|---|---|
| | K70 | G194 | L195 | D196 | F262 | T75 | N190 | D193 | G264 | A321 | H324 | H325 | Q305 | K121 | L122 | L125 | F180 | G184 | N187 | A188 | V296 | T299 | V302 | I303 | D265 | K133 | L191 |
| CAR | | | | | | | | | | | | | | | | | | | | | | | | | | | |
| CAP | | | | | | | | | | | | | | | | | | | | | | | | | | | |
| MUR | | | | | | | | | | | | | | | | | | | | | | | | | | | |
| MD2 | | | | | | | | | | | | | | | | | | | | | | | | | | | |
| TUN | | | | | | | | | | | | | | | | | | | | | | | | | | | |

**Fig. 6** Summary of the hot spots of MraY inhibition. **a** Structural overlay of MraY$_{AA}$ bound to carbacaprazamycin (magenta), capuramycin (yellow), 3′-hydroxymureidomycin A (green), and muraymycin D2 (orange), and MraY$_{CB}$ bound to tunicamycin (slate) viewed from the cytoplasm. **b** Structure of MraY$_{AA}$ in surface representation with inhibitor binding site hot spots (HSs) color-coded and labeled as follows: uridine (red), uridine-adjacent (HS1; lime green), TM9b/LoopE (HS2; purple), caprolactam (HS3; pink), hydrophobic (HS4; cyan), Mg$^{2+}$ (HS5; gold), and tunicamycin (HS6; brown). **c** A barcode representing the interactions each nucleoside inhibitor makes with HS1–6. The residues shown underneath each HS label are found at that site in MraY. Amino acid residue numbering is shown for MraY$_{AA}$ and color-coding is consistent with **b**. Each row represents a different compound: carbacaprazamycin (CAR), capuramycin (CAP), 3′-hydroxymureidomycin A (MUR), muraymycin D2 (MD2), and tunicamycin (TUN). A dark gray square represents an interaction between the corresponding inhibitor and residue. A white square indicates that no contact is made. Squares colored light gray represent that either the amino acid residue side chain or the inhibitor substructure is not resolved in the crystal structure, but likely makes the relevant binding interaction. The side chains of residues K70 and K121 are disordered in the MraY$_{AA}$-capuramycin complex structure. The side chain of residue K70 is disordered in the MraY$_{AA}$-3′-hydroxymureidomycin A complex structure

on the cytosolic surface near the uridine binding site. Our structural studies, in conjunction with previous SAR data, reveal the importance of each druggable hot spot in MraY, and how these sites can be exploited in multiple combinations to maximize the therapeutic potential of MraY-targeted inhibitors.

The uridine-adjacent pocket, HS1, is perhaps the most druggable site on MraY, as this pocket can recognize a surprisingly wide variety of pharmacophores, including 5-aminoribosyl, *meta*-tyrosine, and 3,4-dihydroxy-3,4-dihydro-2*H*-pyran moieties. Because tunicamycin does not extensively utilize this site, targeting it likely improves selectivity for MraY, thereby lessening off-target effects on human GPT that lead to cytotoxicity. Important pharmacophore requirements for HS1 include shape complementarity, the presence of hydrogen bond donor and acceptor functional groups, and perhaps most critically, spatial positioning relative to the uridine moiety. The binding of muraymycin D2 to MraY has been described as akin to a plug inserting into an electrical socket, with the uracil and 5-aminoribosyl moieties serving as the two prongs of the plug[18]. Our structures reveal that the 5-aminoribosyl prong, which binds HS1, can be replaced by a variety of chemical moieties, provided that the geometry of the inhibitor core structure allows for a "plug-like" two-pronged shape that binds both HS1 and the uracil pocket. In muraymycin D2, carbacaprazamycin, and capuramycin, this two-pronged geometry is formed in part by the stereocenter at the 5′ position of the nucleoside ribosyl group (Supplementary Fig. 7). The important role of this chiral center is underscored by two SAR studies of epimeric nucleoside MraY inhibitors. The core structure of muraymycin D2, 5-aminoribosyl uridine, inhibits MraY with 100-fold greater potency if the stereocenter at the 5′ position of the nucleoside ribosyl group is *S*-rather than *R*-configuration[44]. However, a recent study demonstrates that muraymycin analogs lacking the 5-aminoribosyl moiety tolerate either *S*- or *R*-configuration at the 5′ position of the ribosyl group[39]. Collectively, these data are consistent with the notion that the geometry of the nucleoside ribosyl moiety is conducive to forming the two-pronged molecular shape that can bind to the uracil and HS1 pockets on MraY. Interestingly, the stereocenter at the 5′ position of the ribosyl group is only one mechanism by which nature has developed two-pronged inhibitors of MraY. The 5′ position of the ribosyl group in 3′-hydroxymureidomycin A is not a chiral center; there is instead a 4′, 5′-enamide linker at this site. However, the second prong of 3′-hydroxymureidomycin A (*meta*-tyrosine) still accesses HS1 due to the stereochemistry of its core peptidic structure.

HS2, the TM9b/Loop E pocket, can be accessed via interaction with H325 in the Loop E helix, as each nucleoside inhibitor except capuramycin demonstrates. Functionalizing the carboxamide in capuramycin, such as introducing a linker or larger moiety, may improve the affinity of this class of compounds by extending the capuramycin binding site to HS2. More extensive interactions with HS2 are observed in 3′-hydroxymureidomycin A and muraymycin D2 binding, which is primarily achieved via a urea motif found in each compound. These two structures demonstrate that several chemical moieties are tolerated at the amino acid sites in the urea dipeptide motif.

HS3, which binds caprolactam, is a cryptic site uniquely occupied by capuramycin. Due to the sequence variability at and near this site (Supplementary Fig. 6b), this moiety may be functionalized for the design of narrow-spectrum antibiotics. Extensive studies have been conducted to understand and improve upon the inherent antimycobacterial activity of the capuramycins[9,24,45–47]. Among the various capuramycin analogs produced, perhaps the most promising is SQ641, which kills *Mycobacterium tuberculosis* faster than some existing anti-tubercular drugs[24]. Recent studies also show that SQ641

effectively treats *Clostridium difficile* infection[48]. SQ641 differs from capuramycin in that it contains a 2′-*O*-acyl group at the ribosyl moiety and the 7-position of the caprolactam moiety is methylated. The caprolactam binding pocket in MraY$_{AA}$ is composed of K121, L122, and K125. While K121 is highly conserved among MraY orthologs, L122 and K125 are not. The equivalent residues in the MraY ortholog from *M. tuberculosis* H37Rv (Rv2156c) are I106 and S109; therefore, introducing a hydrogen bond acceptor on the caprolactam moiety of SQ641 may improve its antitubercular activity. The MraY ortholog from *C. difficile* contains additional amino acids at this site, which also may be targeted by modifying the caprolactam moiety.

HS4, the hydrophobic binding site on MraY, can accommodate aliphatic chains with widely varying structures. Several naturally occurring liposidomycins have been identified with structural variance in their aliphatic chains[49–51]. For example, liposidomycin Types I and III have a branched lipid tail with ester linkage, while Types II and IV contain a single linear chain[52]. Unsaturation, methylation, and functionalization at various sites on the aliphatic tail moiety are broadly tolerated[6,51]. SAR studies on liposidomycins and the related caprazamycins demonstrate that given a common core structure, varying the aliphatic side chain length does not substantially affect MraY inhibitory activity; however, deacylating the inhibitor altogether leads to decreased activity[6,31,53]. Carbacaprazamycin has a simplified saturated acyl chain at the 3″ position of the diazepanone moiety (Fig. 1a) instead of the more complex aliphatic moieties observed in the liposidomycins and caprazamycins. This modification makes its chemical synthesis less complicated and improves compound stability, while achieving high in vivo potency, with IC$_{50}$ values in the low nanomolar range, and promising activity against *S. aureus*[19,26].

In light of the MraY$_{AA}$-mureidomycin and MraY$_{CB}$-tunicamycin complex structures, targeting HS5, the magnesium-coordinating residue in MraY, is a generalizable strategy for designing MraY nucleoside inhibitors. Although no known nucleoside inhibitors appear to access the other two aspartate residues of the conserved DDD motif required for MraY catalysis, these residues are in close proximity to D265 (Supplementary Fig. 9), and could be targeted.

Our structural findings have been summarized in a barcode system for each MraY nucleoside inhibitor (Fig. 6c), which can be used to design novel MraY inhibitors with improved pharmacological properties. One general and obvious strategy to develop new potent MraY inhibitors may be to introduce additional pharmacophores into existing nucleoside MraY inhibitors in order to capture interactions with additional HSs and to engineer favorable pharmacological properties into MraY-targeted inhibitors. SAR studies of muraymycin analogs with various aliphatic tail moieties[5,28,54] could provide insight into the feasibility of this approach. Such muraymycin analogs likely occupy HS4, the hydrophobic groove, in addition to the interactions muraymycin D2 makes with the uracil pocket, HS1, and HS2. While the already high in vitro inhibitory potency of these muraymycin analogs is not further enhanced by acylation, the in vivo potency of some acylated muramycins is substantially improved, presumably due to increased membrane permeability[5,28,54]. A new strategy is to use the barcode system as a guide to design an MraY inhibitor that targets a novel combination of HSs, which may generate a new type of nucleoside MraY inhibitor with different pharmacological profiles. In principle, a synthetic nucleoside inhibitor targeting all HS1–5 could be developed.

## Methods

**Nanobody screening.** MraY$_{AA}$ nanobodies were raised by phage display technology using immunized llama as a repertoire source in partnership with Creative

Biolabs. MraY$_{AA}$ was purified for llama inoculation according to previously reported methods[17] with some modifications, described as follows. Fractions eluted from cobalt resin containing MraY$_{AA}$ were pooled and heat-treated at 60 °C for 20 min. Contaminating protein precipitated and was pelleted by centrifugation (2900 × g, 5 min). The supernatant was concentrated using 50 kDa molecular weight cutoff centrifugation filters (Millipore) and purified by gel filtration using a Superdex 200 10/300 GL column (GE Healthcare Life Sciences) equilibrated with a buffer containing 20 mM Tris-HCl, 150 mM NaCl, 2 mM dithiothreitol (DTT), 5 mM n-decyl-β-D-maltopyranoside (DM; Anatrace). Peak fractions containing MraY$_{AA}$ were collected, pooled, and the concentration of MraY$_{AA}$ was determined by OD$_{280}$ measurement. Amphipol A8–35 (Anatrace) was added to the protein at a 15× higher concentration than protein. The protein sample containing both detergent and amphipol was incubated at 4 °C with rotation for 4 h. Detergent was removed from the sample using Bio-Beads SM2 (15 mg/mL; Bio-Rad), which were incubated with the protein sample at 4 °C overnight with rotation. The following day, the sample was purified by size exclusion chromatography (SEC) on a Superdex 200 10/300 GL column in phosphate-buffered saline (PBS), pH 7.5. Peak fractions containing MraY$_{AA}$ reconstituted in amphipol were pooled, concentrated to 1 mg/mL and sent to Creative Biolabs for immunization.

Sixty-six unique nanobody sequences were identified and clustered based on their sequence similarity. Representative sequences from each of the clusters, codon-optimized for expression in *Escherichia coli*, were synthesized into an expression vector with a His$_{6×}$ tag and pelB sequence (BioBasic). Of the 23 unique nanobodies, 18 produced protein in a trial expression test. Expression of the 18 nanobodies was then scaled up for protein purification. Nanobody expression plasmids were transformed into C41-DE3 *E. coli* cells, which were used to inoculate 6 L of Terrific Broth (TB; Fischer Scientific). The cultures were incubated with shaking for ~2 h at 37 °C with shaking until an OD$_{600}$ of 0.5 was reached, at which point protein expression was induced with 1 mM IPTG and further incubated at 25 °C overnight (~18 h). Cells were then harvested by centrifugation (6000 × g, 10 min) and resuspended in buffer containing 50 mM Tris-HCl pH 8, 150 mM NaCl, and 20% sucrose. The resuspended cells were rotated for 30 min at room temperature after which they were centrifuged at 13,000 × g for 10 min. The pellet was retained, rapidly resuspended with ice cold buffer (50 mM Tris-HCl pH 8 and 150 mM NaCl), and rotated for 30 min at 4 °C. The sample was then centrifuged (13,000 × g, 10 min) and to the clarified supernatant, 1 mM phenylmethylsulfonyl fluoride (PMSF) and DNase I (20 mg) were added. The mixture was then incubated with cobalt resin (Talon) at 4 °C with rotation for 1 h and the protein was eluted with 200 mM imidazole. Nanobodies were further purified by size exclusion chromatography on a Superdex 200 10/300 GL column equilibrated with 50 mM Tris-HCl pH 8 and 150 mM NaCl.

**Protein purification and crystallization.** The 17 nanobodies that formed a complex with MraY$_{AA}$ were screened in crystallization trials in the presence of inhibitors. To prepare MraY$_{AA}$-nanobody protein complex samples for crystallization, nanobodies were expressed and purified as described above and MraY$_{AA}$ was prepared as reported[17]. Briefly, the gene corresponding to MraY$_{AA}$ was codon-optimized for expression in *E. coli* and synthesized as a fusion with a decahistidine-maltose binding protein (His-MBP) with a PreScission protease site between MraY$_{AA}$ and His-MBP. MraY$_{AA}$ was expressed in C41 (DE3) cells at 37 °C for 4 h. The His-MBP fusion protein was extracted with dodecyl-maltoside (DDM) and purified using a Co$^{2+}$ affinity resin (Talon). His-MBP was cleaved from MraY$_{AA}$ by PreScission protease treatment at 4 °C overnight. MraY$_{AA}$ was combined with each nanobody at a 1:1.5 molar ratio and the complex was purified by SEC with a Superdex 200 10/300 GL column in 20 mM Tris-HCl, 150 mM NaCl, and 5 mM DM. The peak fractions containing the MraY$_{AA}$-nanobody complex were harvested, concentrated to ~450 μM, and combined with capuramycin, carbacaprazamycin, or 3′-hydroxymureidomycin A at 1:1.5–1:3 molar ratio of protein to inhibitor. All MraY$_{AA}$-nanobody-inhibitor complexes were screened for crystallization via sitting drop vapor diffusion using MemGold™ (Molecular Dimensions) and in-house crystallization screening solutions. Of the 17 nanobody complexes screened, 15 produced crystals that were tested for diffraction. One nanobody in particular, NB7, produced the best diffracting crystals in the presence of each of the inhibitors tested. For the MraY$_{AA}$-NB7-carbacaprazamycin complex, crystals formed at 17 °C in 20% polyethelyene glycol (PEG) 4000, 0.2 M potassium thiocyanate, 0.1 M sodium acetate pH 4.6. For the MraY$_{AA}$-NB7-capuramycin complex, crystals formed at 17 °C in 18% PEG 4000, 0.4 M ammonium thiocyanate, 0.1 M sodium acetate pH 4.6. For the MraY$_{AA}$-NB7–3′-hydroxymureidomycin A complex, crystals formed at 17 °C in 20% PEG 4000, 0.2 M ammonium thiocyanate, 0.1 M sodium acetate pH 4.6. All crystals were equilibrated to 4 °C for 24 h prior to harvesting and flash cooling.

**Data collection and structure determination.** X-ray crystal diffraction data were collected on the SERCAT 22-ID and NECAT 24-IDC and 24-ID-E beamlines (Advanced Photo Source, Argonne National Laboratory) using a wavelength of 1.00 or 0.979 Å. All datasets were processed with XDS[55]. For each inhibitor-bound MraY$_{AA}$ structure, datasets from multiple isomorphous crystals were merged using BLEND[56]. For the merged data of MraY$_{AA}$ in complex with 3′-hydroxymureidomycin A, diffraction anisotropy was corrected by ellipsoidal truncation using the STARANISO server[57]. Phasing for each structure was obtained by

molecular replacement in PHASER[58] using as search models: (1) the structure of MraY$_{AA}$-muraymycin D2 (PDB ID: 5CKR) with the inhibitor, TM9b, Loop E, and the Loop E helix removed, and (2) a high-resolution structure of a nanobody deposited in the Protein Data Bank (PDB ID: 4C57). The crystals obtained of MraY$_{AA}$ in complex with NB7 and each inhibitor were in the P21 space group with two MraY$_{AA}$ dimers and 4 NB7 molecules in the asymmetric unit. Inhibitor density was strongest in one MraY$_{AA}$ protomer in each structure, probably owing to nanobody crystal packing at this site. Manual model building was performed in COOT[59] and refinement in PHENIX.refine[60]. For the initial molecular replacement solution of MraY$_{AA}$ in complex with 3′-hydroxymureidomycin A, jelly-body refinement was first performed using LORESTR[61]. Molecular graphics were generated using PyMOL[62]. For the MraY$_{AA}$-capuramycin structure, sequence conservation was mapped onto the protein surface using the ConSurf server[63] with 30 MraY orthologs sequences for the alignment. Data collection and refinement statistics are provided in Table 1.

**UMP-Glo assay.** The UMP-Glo™ glycosyltransferase assay[64] was carried out according to the manufacturer′s specifications (Promega Corporation). For both IC$_{50}$ and specific activity measurements, reaction mixtures contained 150 μM UDP-MurNAc-pentapeptide (UM5A) and 250 μM undecaprenyl phosphate (C$_{55}$-P) in a buffer composed of 100 mM Tris-HCl, 500 mM NaCl, 10 mM MgCl$_2$, and 20 mM (3-((3-cholamidopropyl) dimethylammonio)−1-propanesulfonate) (CHAPS; Anatrace). The reaction was initiated with MraY$_{AA}$ to a final concentration of 50 nM. Reactions were carried out for 5 min at 45 °C. All luminescence measurements were normalized relative to a negative control reaction without enzyme. For IC$_{50}$ measurements, the following concentrations were used. Carbacaprazamycin: 0.01, 0.05, 0.1, 0.5, 1, 20, and 220 μM; capuramycin: 0.01, 0.1, 0.5, 1, 2.5, 50, and 375 μM; 3′-hydroxymureidomycin A: 0.01, 0.05, 0.1, 1, 5, 50, and 370 μM. For specific activity measurements, NB7 and each inhibitor were added to a final concentration of 1 μM and 0.5 μM where present. Luminescence measurements were made using a SpectraMax M3 multi-mode microplate reader.

**Synthesis of capuramycin.** Capuramycin was synthesized according to the known procedure[65]. β-Uridine was partially protected with BOM and Tr groups to yield 3-((benzyloxy)methyl)-1-((2R,3R,4S,5R)-3,4-dihydroxy-5-((trityloxy)methyl)tetrahydrofuran-2-yl)pyrimidine-2,4(1H,3H)-dione. Conversion of the above intermediate to the corresponding 2-O-acetyl-3-O-methyl-uridine derivative was achieved through mono-methylation, acetylation, and detritylation. Dess–Martin oxidation of the 2-O-acetyl-3-O-methyl-uridine derivative followed by addition of TMSCN to the resulting aldehyde afforded the cyanohydrin ((2R,3R,4R,5R)-2-(3-((benzyloxy)methyl)-2,4-dioxo-3,4-dihydropyrimidin-1(2H)-yl)-5-((S)-cyano(hydroxy)methyl)-4-methoxytetrahydrofuran-3-yl acetate). Coupling of the cyanohydrin and tetraacetyl thio-α-D-mannopyranoside followed by hydrolysis gave the corresponding amide. Selective deacetylation of the primary acetate using I$_2$ and removal of the uracil BOM group of the amide provided the primary alcohol ((2S,3S,4S)-2-((R)-1-((2S,3R,4R,5R)-4-acetoxy-5-(2,4-dioxo-3,4-dihydropyrimidin-1(2H)-yl)-3-methoxytetrahydrofuran-2-yl)-2-amino-2-oxoethoxy)-6-(hydroxymethyl)-3,4-dihydro-2H-pyran-3,4-diyl diacetate). Parikh–Doering oxidation of the primary alcohol to the α,β-unsaturated aldehyde followed by Pinnick oxidation afforded the corresponding carboxylic acid ((2S,3S,4S)-3,4-diacetoxy-2-((R)-1-((2S,3R,4R,5R)-4-acetoxy-5-(2,4-dioxo-3,4-dihydropyrimidin-1(2H)-yl)-3-methoxytetrahydrofuran-2-yl)-2-amino-2-oxoethoxy)-3,4-dihydro-2H-pyran-6-carboxylic acid). Final coupling of the carboxylic acid with 2-(S)-aminocaprolactam and exhaustive deprotection accomplished the synthesis of capuramycin.

**Synthesis of carbacaprazamycin.** Carbacaprazamycin was synthesized according to a procedure previously reported[26]. A mixture of (2S,3R)-tert-butyl-3-hydroxymethyl-2-[N-methyl-(1-phenylfluorenyl)amino]hex-5-enoate and 37% aqueous HCHO in THF was irradiated at 150 °C (9 bar). The mixture was concentrated in vacuo, and the residue was purified by silica gel column chromatography to afford (4S,5R)-4-tert-butoxycarbonyl-3-(1-phenylfluorenyl)-1,3-oxadinane. A solution of this compound and AcOH in CH$_2$Cl$_2$ was stirred at room temperature for 15 min. Sodium triacetoxyborohydride was then added to the mixture, which was stirred at room temperature. The mixture was diluted with AcOEt and washed with saturate aqueous NaHCO$_3$ and saturate aqueous NaCl. The organic layers was dried with Na$_2$SO$_4$, filtered, and concentrated in vacuo, and the residue was purified by silica gel column chromatography to afford (2-S,3R)-tert-butyl 3-hydroxymethyl-2-[N-methyl-(1-phenylfluorenyl)amino]hex-5-enoate. A mixture of this compound, hexadecene, and Gubbs second catalyst in CH$_2$Cl$_2$ was heated under reflux. The mixture was cooled to room temperature and concentrated in vacuo. The residue was passed through a silica gel pad with 50% AcOEt in hexane as an eluent to give a crude heneicosanate, which was used to the next step. A mixture of the heneicosanate, AcOH and Pd(OH)$_2$ in MeOH was vigorously stirred under H$_2$ atmosphere at room temperature. The catalyst was filtered off through a Celite pad, and the filtrate was concentrated in vacuo. The residue was purified by silica gel column chromatography (75% AcOEt–hexane) to afford (2S,3R)-tert-butyl 3-hydroxymethyl-2-N-methylaminoheneicosanate. A solution of this compoundand imidazole in CH$_2$Cl$_2$ was treated with TBSCl at room temperature. Few drops of MeOH was added to the mixture, which was further stirred for 5 min. The mixture

was diluted with AcOEt, which was washed with 0.1 M aqueous HCl, saturate aqueous NaHCO$_3$ and saturate aqueous NaCl. The organic layers were dried with Na$_2$SO$_4$, filtered, and concentrated in vacuo to give a crude amine. A mixture of the crude amine and 6-benzyloxycarbonylamino-5-O-[5-tert-butoxycarbonylamino-5-deoxy-2,3-O-(3-pentylidene)-β-D-ribo-pentofuranosyl]-6-deoxy-2,3-O-isopropylidene-1-(uracil-1-yl)-β-D-glycelo-L-talo-heptofuranuronate in THF was treated sequentially with NaHCO$_3$ and DEPBT at 0 °C, which was allowed to room temperature. The reaction mixture was partitioned between AcOEt and saturated aqueous NaHCO$_3$. The organic phase was washed with saturated aqueous NaCl, dried with Na$_2$SO$_4$, filtered, and concentrated in vacuo. The residue was purified by silica gel column chromatography to afford N-[(1S,2R)-1-tert-butoxycarbonyl-2-tert-butyldimethylsilyloxymethyleicosanyl]-N-methyl-6-benzyloxycarbonylamino-1-(3-benzyloxymethyluracil-1-yl)-5-O-[5-tert-butoxycarbonylamino-5-deoxy-2,3-O-(3-pentylidene)-β-D-ribofuranosyl]-6-deoxy-2,3-O-isopropylidene-β-D-glycero-L-talo-heptofuranuronamide. A solution of this compound in MeCN was treated with 3HF·Et$_3$N at room temperature. The mixture was diluted with AcOEt, which was washed with saturated aqueous NaCl and saturated aqueous NaCl, dried (Na$_2$SO$_4$), filtered, and concentrated in vacuo to give a crude alcohol. A solution of the alcohol in CH$_2$Cl$_2$ was treated with Dess–Martin periodinane at room temperature. The mixture was diluted with AcOEt, and a mixture of saturated aqueous NaHCO$_3$ and saturated aqueous Na$_2$S$_2$O$_3$ was added. The whole mixture was vigorously stirred, and the organic phase was dried (Na$_2$SO$_4$), filtered, and concentrated in vacuo to give a crude aldehyde. A mixture of the aldehyde and Pd black in i-PrOH was vigorously stirred under a H$_2$ atmosphere at room temperature. The catalyst was filtered off through a Celite pad, and the filtrate was concentrated in vacuo. The residue in CH$_2$Cl$_2$ was treated with AcOH and NaBH (OAc)$_3$, and the reaction mixture was stirred at room temperature. The mixture was partitioned between AcOEt and saturated aqueous NaHCO$_3$. The organic phase was washed with saturated aqueous NaCl, dried (Na$_2$SO$_4$), filtered, and concentrated in vacuo. The residue was purified by silica gel column chromatography to afford a white foam. A solution of this material in 80% aqueous TFA was stirred at room temperature. The volatiles were removed in vacuo to afford carbacaprazamycin.

**Reporting summary.** Further information on research design is available in the Nature Research Reporting Summary linked to this article.

## Data availability
Data supporting the findings of this manuscript are available from the corresponding author upon reasonable request. The source data underlying Supplementary Figs. 1b, 2b are provided as a Source Data file. Atomic coordinates and structure factors for the reported crystal structures are deposited in the Protein Data Bank under accession codes 6OYH, 6OYZ, and 6OZ6.

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

## Acknowledgements

Data for this study were collected at beamlines NECAT 24-ID-C and 24-ID-E and at SERCAT 22-ID, both at the Advanced Photon Source. We thank A. Kuk and N. Wright for help with data processing and model building and J. Yoo for help with initial nanobody screening. This work was supported by the National Institutes of Health (R01GM120594 to S.-Y.L.), JSPS Grant-in-Aid for Scientific Research (16H05097, 18H04599, and 19H03345 to S.I.), Astellas Foundation for Research on Metabolic Disorders (to S.I.), Hokkaido University GFC, PSOU, funded by MEXT (to S.I.), and BINDS from the Japan Agency for Medical Research and Development (to S.I.). Beamlines 24-ID-C and 24-ID-E are funded by P30 GM124165 and S10 RR029205.

## Author contributions

E.H.M. performed nanobody screening, MraY crystallization, data collection, model building, and enzyme assays; B.K. performed nanobody screening, MraY crystallization, and assisted in data collection and enzyme assay, all under the guidance of S.-Y.L. Y.T. and A.K. synthesized carbacaprazamycin and 3′-hydroxymureidomycin A under the guidance of S.I. D.-Y.K. and K.L. synthesized capuramycin under the guidance of J.H. E. H.M. and S.-Y.L. wrote the paper with input from the rest of the authors.

## Additional information

**Competing interests:** The authors declare no competing interests.

