## [Peer Review File · Nature Communications]

Reviewers' comments:

Reviewer #1 (Remarks to the Author):

The paper by Lee and colleagues describes X-ray crystal structures of the bacterial enzyme *MraY* (a promising unexploited antibacterial drug target) in complex with three structurally distinct uridine-derived inhibitors, representing three different classes of so-called 'nucleoside antibiotics'. These X-ray crystal structures surely were challenging to obtain as *MraY* is an integral membrane protein. The careful analysis of the three novel as well as two previously reported crystal structures of *MraY*-inhibitor complexes furnished a comprehensive model for *MraY* inhibition by uridine-derived antibiotics, based on the identification of six hot spots besides the uridine binding pocket.

This is some excellent work with high relevance for the future development of *MraY*-inhibiting antibiotics, and it therefore represents a milestone in this field of antimicrobial research. The manuscript is well written, clear, and conclusive. I therefore wholeheartedly vote for publication of the paper in *Nature Communications*. However, there are some minor issues which should be addressed by the authors prior to publication of their work.

Mainly, I would like to challenge one aspect in their conclusions in the "Discussion" section. The authors suggest that more potent *MraY* inhibitors might be accessible if more interactions with the identified hotspots are captured. They demonstrate this hypothesis by referring to some previous results obtained for the sub-class of muraymycins. However, the data presented in reference 5 might suggest that this approach is not necessarily successful. Thus, muraymycins of the B-series and C-series had been reported in the referenced work to display similar (very strong) inhibitory potencies towards *MraY* in the pM range. However, B-series muraymycins are lipophilically acylated and C-series muraymycins lack lipophilic decoration. Based on the hotspot model introduced by the authors, one would assume B-series muraymycins to be the stronger *MraY* inhibitors though as they apparently exploit interactions with hotspot 4 (in contrast to C-series congeners). This contradiction might be explained by the already very high inhibitory potency of C-series muraymycins - maybe the exploitation of additional hotspots does not necessarily provide advantages in such a case. Overall, I think that the authors should somehow include this aspect in their discussion - it does not diminish the high quality of their results, but may provide an interesting aspect to their proposal.

There are some further minor issues which should also be addressed:

1) Page 11: The authors state that "the stereocenter at the 5' position of the ribosyl moiety must be S-configuration (sic!) for efficient inhibitory activity of *MraY*". Maybe they are not aware of a more recent publication which strongly suggests otherwise: Spork et al., Analogues of Muraymycin Nucleoside Antibiotics with Epimeric Uridine-Derived Core Structures, *Molecules* 2018, 23, 2868.

2) Page 11: The authors suggest that the replacement of the ribose moiety "could simplify the chemical synthesis of some nucleoside *MraY* inhibitors". I would like to challenge this statement as nearly all syntheses of uridine-derived nucleoside antibiotics start from uridine. In uridine, however, the uracil base is already connected to the ribose unit with the correct stereochemistry, which would be needed to be achieved otherwise in the analogues suggested by the authors. Hence, such a replacement would make the synthesis of corresponding *MraY* inhibitors probably more difficult rather than more facile.

3) Page 13: The authors state that "occupying the uridine-adjacent pocket is not required for *MraY* inhibition". I agree with this conclusion, but the authors might want to add that this is also supported by the reported *MraY*-inhibiting activities of naturally occurring muraymycins A5 and C4 (which both lack the 5'-aminoribosyl moiety filling the uridine-adjacent pocket) as well as some synthetic 5'-defunctionalised analogues (also lacking this aminoribose unit, see reference 5 as well as these two publications: Spork et al., Lead Structures for New Antibacterials: Stereocontrolled Synthesis of a Bioactive Muraymycin Analogue, *Chem. Eur. J.* 2014, 20, 15292-15297; Spork et al., Analogues of Muraymycin Nucleoside Antibiotics with Epimeric Uridine-Derived Core Structures, *Molecules* 2018, 23, 2868 (already mentioned above)).

4) Nomenclature: Throughout the paper, the authors refer to an "amino-urea-amino" motif found in muraymycins and mureidomycins. This terminology is very misleading, in my opinion - it is probably meant that it is a urea unit formed by two amino acids, but this is not correctly

represented in this terminology. I would therefore advise to replace "amino-urea-amino" with "urea dipeptide" throughout the manuscript.

5) Figure 1: There is a typo here ("dizapanone").

6) References: Reference 29 is incomplete, and standard journal title abbreviations should be used throughout.

7) Supplementary Figure 4: The chemical structure of muraymycin D2 is not correct as one methylene unit is missing.

Reviewer #2 (Remarks to the Author):

In manuscript, Mashalidis E et al. reported three crystal structures of *MraY* in complex with caprazamycin, capuramycin, and 3'-hydroxymureidomycin A, three representative members of nucleoside inhibitors of *MraY*. These structures revealed cryptic druggable hot spots in the shallow inhibitor binding site of *MraY* and critical residues of *MraY* involved in interactions with inhibitors. Although these structures do not represent the first reported structures of *MraY* or *MraY* in complex with an inhibitor, this study does provide interesting insights into the chemical logic of *MraY* inhibition. As *MraY* is highly conserved among Gram-positive and Gram-negative bacteria, the structure information could guide novel approaches to *MraY*-targeted antibiotic design. In general, this manuscript is well written and easy to follow. I recommend it for publication with minor changes.

Minor changes:

1. Page 27, Figure 6. "(b) A barcode representing the interactions each nucleoside inhibitor makes with HS1-6." should change to "(c) A barcode representing the interactions each nucleoside inhibitor makes with HS1-6."

Point-by-point response to the reviewers' comments

Reviewer #1

Major point:

*Mainly, I would like to challenge one aspect in their conclusions in the "Discussion" section. The authors suggest that more potent *MraY* inhibitors might be accessible if more interactions with the identified hotspots are captured. They demonstrate this hypothesis by referring to some previous results obtained for the sub-class of muraymycins. However, the data presented in reference 5 might suggest that this approach is not necessarily successful. Thus, muraymycins of the B-series and C-series had been reported in the referenced work to display similar (very strong) inhibitory potencies towards *MraY* in the pM range. However, B-series muraymycins are lipophilically acylated and C-series muraymycins lack lipophilic decoration. Based on the hotspot model introduced by the authors, one would assume B-series muraymycins to be the stronger *MraY* inhibitors though as they apparently exploit interactions with hotspot 4 (in contrast to C-series congeners). This contradiction might be explained by the already very high inhibitory potency of C-series muraymycins - maybe the exploitation of additional hotspots does not necessarily provide advantages in such a case. Overall, I think that the authors should somehow include this aspect in their discussion - it does not diminish the high quality of their results, but may provide an interesting aspect to their proposal.*

We thank the reviewer for raising this issue and agree the example of the acylated muraymycin analogs requires further explanation. As such, we have amended the text in the Discussion section (p. 20):

“One general and obvious strategy to develop new potent *MraY* inhibitors may be to introduce additional pharmacophores into existing nucleoside *MraY* inhibitors in order to capture interactions with additional HSs and to engineer favorable pharmacological properties into *MraY*-targeted inhibitors. SAR studies of muraymycin analogs with various aliphatic tail moieties (Tanino 2010, Tanino 2011, Kopperman 2018) could provide insight into the feasibility of this approach. Such muraymycin analogs likely occupy HS4, the hydrophobic groove, in addition to the interactions muraymycin D2 makes with the uracil pocket, HS1, and HS2. While the already high *in vitro* inhibitory potency of these muraymycin analogs is not further enhanced by acylation, the *in vivo* potency of some acylated muraymycins is substantially improved, presumably due to increased membrane permeability (Tanino 2010, Tanino 2011, Kopperman 2018).”

Minor points:

1) Page 11: The authors state that "*the stereocenter at the 5' position of the ribosyl moiety must be S-configuration (sic!) for efficient inhibitory activity of *MraY**". Maybe they are not aware of a more recent publication which strongly suggests otherwise: Spork et al., Analogues of

We appreciate the reviewer's point and offer the following clarification. Our structural analyses are consistent with both the data demonstrating the importance of *S*-configuration at the 5' position of the ribosyl moiety (Dini 2000) and the more recent epimer study of muraymycin derivatives (Spork 2018). To explain this further, we have removed analysis of the stereochemistry from the subsection entitled "The uridine binding site is common to *MraY* nucleoside inhibitors" and added the following text to the Discussion (p. 17):

"The binding of muraymycin D2 to *MraY* has been described as akin to a plug inserting into an electrical socket, with the uracil and 5-aminoribosyl moieties serving as the two prongs of the plug (Chung 2016). Our structures reveal that the 5-aminoribosyl prong, which binds HS1, can be replaced by a variety of chemical moieties, provided that the geometry of the inhibitor core structure allows for a "plug-like" two-pronged shape that binds both HS1 and the uracil pocket. In muraymycin D2, carbacaprazamycin, and capuramycin, this two-pronged geometry is formed in part by the stereocenter at the 5' position of the nucleoside ribosyl group (Supplementary Figure 7). The important role of this chiral center is underscored by two SAR studies of epimeric nucleoside *MraY* inhibitors. The core structure of muraymycin D2, 5-aminoribosyl uridine, inhibits *MraY* with 100-fold greater potency if the stereocenter at the 5' position of the nucleoside ribosyl group is *S*- rather than *R*-configuration (Dini 2000). However, a recent study demonstrates that muraymycin analogs lacking the 5-aminoribosyl moiety tolerate either *S*- or *R*-configuration at the 5' position of the ribosyl group (Spork 2018). Collectively, these data are consistent with the notion that the geometry of the nucleoside ribosyl moiety is conducive to forming the two-pronged molecular shape that can bind to the uracil and HS1 pockets on *MraY*. Interestingly, the stereocenter at the 5' position of the ribosyl group is only one mechanism by which nature has developed two-pronged inhibitors of *MraY*. The 5' position of the ribosyl group in 3'-hydroxymureidomycin A is not a chiral center; there is instead a 4', 5'-enamide linker at this site. However, the second prong of 3'-hydroxymureidomycin A (*meta*-tyrosine) still accesses HS1 due to the stereochemistry of its core peptidic structure."

2) Page 11: *The authors suggest that the replacement of the ribose moiety "could simplify the chemical synthesis of some nucleoside MraY inhibitors". I would like to challenge this statement as nearly all syntheses of uridine-derived nucleoside antibiotics start from uridine. In uridine, however, the uracil base is already connected to the ribose unit with the correct stereochemistry, which would be needed to be achieved otherwise in the analogues suggested by the authors. Hence, such a replacement would make the synthesis of corresponding MraY inhibitors probably more difficult rather than more facile.*

We have now removed the speculation that such a modification to the ribosyl moiety would make the synthesis more facile.

3) Page 13: *The authors state that "occupying the uridine-adjacent pocket is not required for MraY inhibition". I agree with this conclusion, but the authors might want to add that this is also supported by the reported MraY-inhibiting activities of naturally occurring muraymycins A5*

and C4 (which both lack the 5'-aminoribosyl moiety filling the uridine-adjacent pocket) as well as some synthetic 5'-defunctionalised analogues (also lacking this aminoribose unit, see reference 5 as well as these two publications: Spork et al., *Lead Structures for New Antibacterials: Stereocontrolled Synthesis of a Bioactive Muraymycin Analogue*, *Chem. Eur. J.* 2014, 20, 15292-15297; Spork et al., *Analogues of Muraymycin Nucleoside Antibiotics with Epimeric Uridine-Derived Core Structures*, *Molecules* 2018, 23, 2868 (already mentioned above)).

We have added these citations to strengthen our conclusion that the uridine-adjacent pocket is not required for MraY inhibition. On p. 13, paragraph 1, lines 7-8, we have amended the text as follows:

“Occupying the uridine-adjacent pocket is not required for MraY inhibition, but it enhances inhibitory potency. This observation is bolstered by SAR studies demonstrating that muraymycin analogs lacking the 5-aminoribosyl moiety that binds the uridine-adjacent site, such as muraymycins A5 and C4 and some synthetic 5'-defunctionalized muraymycin derivatives, retain inhibitory activity (McDonald 2002; Spork 2014; Spork 2018).”

4) *Nomenclature: Throughout the paper, the authors refer to an "amino-urea-amino" motif found in muraymycins and mureidomycins. This terminology is very misleading, in my opinion - it is probably meant that it is a urea unit formed by two amino acids, but this is not correctly represented in this terminology. I would therefore advise to replace "amino-urea-amino" with "urea dipeptide" throughout the manuscript.*

We have replaced the term “amino-urea-amino” with “urea dipeptide” throughout the text, figures, and figure legends.

5) *Figure 1: There is a typo here ("dizapanone").*

The spelling has been changed to “diazepanone” in Figure 1.

6) *References: Reference 29 is incomplete, and standard journal title abbreviations should be used throughout.*

We have provided the full reference details for #29 and have corrected the title abbreviations for all references.

7) *Supplementary Figure 4: The chemical structure of muraymycin D2 is not correct as one methylene unit is missing.*

We have added a methylene unit to the 2D chemical structure of muraymycin D2 in Supplementary Figure 4a.

Reviewer #2

Minor changes:

1. Page 27, Figure 6. “ (b) A barcode representing the interactions each nucleoside inhibitor makes with HSI-6.” should change to “ (c) A barcode representing the interactions each nucleoside inhibitor makes with HSI-6.”

We have now corrected the figure legend for Figure 6 as described above.